# DiScoFormer: Plug-In Density and Score Estimation with Transformers

**Vasily Ilin**[*]
Department of Mathematics
University of Washington
Seattle, USA
vilin@uw.edu

**Peter Sushko**
Allen Institute for Artificial Intelligence
Seattle, USA

## Abstract

Estimating probability density and its score from samples remains a core problem in generative modeling, Bayesian inference, and kinetic theory. Existing methods are bifurcated: classical kernel density estimators (KDE) generalize across distributions but suffer from the curse of dimensionality, while neural score-matching models achieve high precision but require retraining for every target distribution. We introduce DiScoFormer (Density and Score Transformer), an equivariant Transformer that maps i.i.d. samples to both density values and score vectors. Unlike score matching, which learns a fixed function $\mathbb{R}^d \to \mathbb{R}^d$ for a single distribution, DiScoFormer learns a *sequence-to-sequence* operator that generalizes across distributions and sample sizes without retraining. Analytically, we prove that self-attention can recover normalized KDE, establishing it as a functional generalization of kernel methods; empirically, individual attention heads learn multi-scale, kernel-like behaviors. The model outperforms KDE for density and score estimation, and provides a plug-in score oracle for score-debiased KDE, Fisher information computation, and Fokker–Planck-type PDEs.

## 1 Introduction

Estimating a probability density $f$ and its score $\nabla \log f$ from i.i.d. samples is foundational in generative modeling, Bayesian inference, and kinetic theory. Classical KDE offers strong theoretical guarantees but suffers from the curse of dimensionality, while neural score-matching models achieve high accuracy but require retraining for every target distribution.

We introduce DiScoFormer (Density and Score Transformer), an equivariant Transformer that maps i.i.d. samples to both density values and score vectors. Specifically, we learn operators $T$ and $S$:

$$\begin{pmatrix} x_1 \\ \vdots \\ x_n \end{pmatrix} \xrightarrow{T} \begin{pmatrix} f(x_1) \\ \vdots \\ f(x_n) \end{pmatrix}, \qquad \begin{pmatrix} x_1 \\ \vdots \\ x_n \end{pmatrix} \xrightarrow{S} \begin{pmatrix} \nabla \log f(x_1) \\ \vdots \\ \nabla \log f(x_n) \end{pmatrix}.$$

**A different task from score matching.** Score matching (Hyvärinen, 2005) and its variants (sliced (Song et al., 2019), denoising (Vincent, 2011)) learn a function $s_\theta \colon \mathbb{R}^d \to \mathbb{R}^d$ that approximates $\nabla \log f$ for a *single, fixed* distribution; the model must be retrained from scratch whenever the target changes. DiScoFormer solves a fundamentally different problem: it learns operators $T, S$ that map *sequences* in $\mathbb{R}^d$ to sequences in $\mathbb{R}^1$ and $\mathbb{R}^d$, respectively, so that a single forward pass estimates the density and score of *whatever distribution* produced the input samples. This sequence-to-sequence formulation enables applications where per-distribution retraining is impractical, such as particle methods that change the underlying density at every timestep.

These operators must respect permutation and affine equivariance: $T(PXA + \mathbf{1}\mu^\top) = P \,|\det A|^{-1} T(X)$ and $S(PXA + \mathbf{1}\mu^\top) = P A^{-1} S(X)$ for permutation matrix $P$, invertible $A$, and shift $\mu$. The Transformer architecture provides permutation equivariance by construction;

---

[*]Corresponding author: vilin@uw.edu

we achieve affine equivariance through whitening combined with data augmentation for rotation invariance.

We prove that self-attention can recover normalized KDE weights, establishing Transformers as data-adaptive generalizations of kernel methods. Empirically, the model outperforms KDE across dimensions and sample sizes on both in-distribution and out-of-distribution targets, and provides plug-in oracles for downstream tasks including entropy estimation, Fisher information calculation, and Fokker–Planck-type PDEs.

**Contributions:** (1) DiScoFormer, a distribution-agnostic Transformer for amortized density and score estimation; (2) A proof that self-attention recovers normalized KDE; (3) Superior accuracy to KDE across sample sizes and dimensions; (4) Applications to Fisher information, entropy, and Fokker–Planck PDEs.

## 2   RELATED WORK

**Kernel density estimation.**   Classical KDE methods (Parzen, 1962; Silverman, 1986) estimate densities via local kernel averaging but suffer from a rigid bias–variance trade-off and scale poorly with dimension (Scott, 2015). Score-Debiased KDE (Epstein et al., 2025) shows that access to a score oracle enables bias correction—we provide precisely this oracle as a reusable, distribution-agnostic component.

**Score matching and generative models.**   Score matching (Hyvärinen, 2005; Vincent, 2011), its sliced variant (Song et al., 2019), and score-based generative models (Song & Ermon, 2019; Ho et al., 2020; Song et al., 2021) learn a function $s_\theta \colon \mathbb{R}^d \to \mathbb{R}^d$ for a single target density. In contrast, our model learns a sequence-to-sequence operator that maps samples to scores across distributions without retraining—a fundamentally different task suited to settings where the target density changes (e.g., particle methods). We provide a qualitative comparison with sliced score matching in Appendix G.

**Permutation-invariant architectures.**   DeepSets (Zaheer et al., 2017), Set Transformers (Lee et al., 2019), and Neural Processes (Garnelo et al., 2018) provide foundations for exchangeable data. Our architecture outputs density and score while additionally enforcing affine equivariance.

**Attention as kernel smoothing.**   Connections between attention and kernels have been explored via linear-attention (Katharopoulos et al., 2020), random features (Choromanski et al., 2021), and Nadaraya–Watson interpretations (Zhang et al., 2023). Proposition 3.2 shows that self-attention can exactly reproduce normalized Gaussian KDE weights.

**Particle methods.**   Interacting particle methods for Fokker–Planck equations (Carrillo et al., 2019; Boffi & Vanden-Eijnden, 2023; Ilin et al., 2025) require accurate score estimates. Our model provides a fast, pretrained oracle that plugs directly into these solvers.

## 3   METHODOLOGY

In this section we describe the core ideas: the symmetry-aware architecture, the relationship to classical KDE, and the training pipeline.

### 3.1   EQUIVARIANCE

Proposition 3.1 establishes the equivariance properties of density and score estimation. The proof is in Appendix C.

**Proposition 3.1** (Permutation and affine equivariance)**.** *Let $f$ be a differentiable density, and $X = (x_1, \dots, x_n)^T$ be its iid sample. Define $T(X) := (f(x_1), \dots, f(x_n))^T$ and $S(X) := (\nabla \log f(x_1), \dots, \nabla \log f(x_n))^T$. Let $P \in \mathbb{R}^{n \times n}$ be a permutation matrix, $A \in \mathbb{R}^{d \times d}$ be invertible, $\mu \in \mathbb{R}^d$, and $\mathbf{1} \in \mathbb{R}^n$ be the vector of ones. Then*

$$T\big(PXA + \mathbf{1}\mu^\top\big) = P\,|\det A|^{-1}\,T(X),$$
$$S\big(PXA + \mathbf{1}\mu^\top\big) = P\,A^{-1}\,S(X).$$

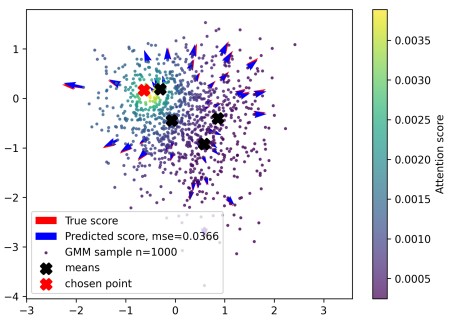

Figure 1: Attention visualization: average attention scores in layer 0 with respect to the chosen point x. The Transformer learns to attend to nearby points.

Table 1: Relative MSE of affine equivariance error (50 trials).

| Transform | Rel. MSE |
|---|---|
| Permutation | 0 |
| Translation | 0 |
| Isotropic scaling | 0 |
| Anisotropic scaling | 0 |
| Rotation | $5 \times 10^{-4}$ |
| Full affine | $1 \times 10^{-4}$ |

To capture these symmetries, we use the Transformer architecture without positional encodings, combined with an affine normalization layer (see Appendix A). The whitening step yields equivariance up to rigid rotations $O(d)$; training on randomly oriented GMMs encourages approximate rotation invariance. Table 1 confirms this empirically.

### 3.2 ATTENTION AND KDE

Proposition 3.2 demonstrates that a single attention head can compute the classical KDE, showing that attention has the capacity to implement kernel-based smoothing.

**Proposition 3.2** (Attention can compute KDE). *Let $X \in \mathbb{R}^{n \times d}$ with $\|x_i\| = 1$ for all $i$. Consider self-attention with $Q = K = \frac{1}{h}X$ and $V = I_n$ for bandwidth $h > 0$. Then*

$$[\text{Attention}(Q, K, V)]_{ij} = \frac{\exp\left(-\frac{\|x_i - x_j\|^2}{2h^2}\right)}{\sum_{k=1}^{n} \exp\left(-\frac{\|x_i - x_k\|^2}{2h^2}\right)},$$

*i.e. the attention matrix coincides with the normalized Gaussian kernel of bandwidth $h$.*

Empirically, the Transformer learns to attend to nearby points (Figure 1), with emergent head specialization where different heads attend at different scales and directions (see Appendix F).

### 3.3 TRAINING AND ARCHITECTURE

We train the Transformer on Gaussian Mixture Models (GMMs), chosen because they form a dense family in the space of continuously differentiable densities with closed-form expressions. The objective combines MSE of predicted density and score: $\mathcal{L} = \alpha \mathcal{L}_T + (1 - \alpha)\mathcal{L}_S$.

The architecture consists of a linear embedding, four Transformer encoder layers (128-dimensional, 8 heads, GELU, pre-norm), and linear projection heads. We use cross-attention to enable evaluation at arbitrary query points $Y$ beyond the observed samples $X$. The model has $\sim$800K parameters. See Appendix A for details.

## 4 EXPERIMENTS

We empirically investigate comparison with KDE and SD-KDE Epstein et al. (2025), scaling in $n$ and $d$, out-of-distribution generalization, and downstream applications. All experiments use a single 48GB L40S GPU. Additional experiments and visualizations are in the Appendix: score estimation details and qualitative plots (Appendix D), a comparison with sliced score matching (Appendix G), runtime scaling (Appendix H), entropy and Fisher information (Appendix D.3), relative Fisher information (Appendix E), and attention head analysis (Appendix F).

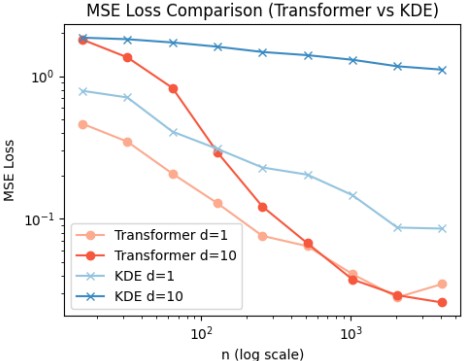 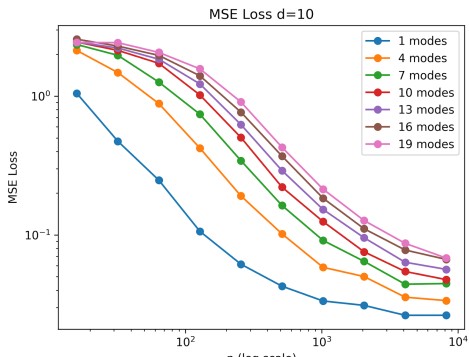

Figure 2: Left: MSE of score estimation in dimensions 1 and 10 on a 3-modal GMM. The Transformer has excellent scaling in both $d$ and $n$. Right: Generalization to GMMs with more modes than seen during training.

Table 2: MSE of score estimation on the 2D Laplace distribution.

| $n$ | KDE | Transformer |
|---|---|---|
| 512 | 0.3810 | **0.3598** |
| 1024 | 0.3305 | **0.2992** |
| 2048 | 0.2990 | **0.2756** |
| 4096 | 0.2650 | **0.2597** |

Table 3: MSE of score estimation on the 2D Student-$t$ distribution ($\nu = 3$).

| $n$ | KDE | No TTT | TTT 4 |
|---|---|---|---|
| 128 | 0.152 | 0.198 | **0.145** |
| 256 | 0.121 | 0.112 | **0.090** |
| 512 | 0.092 | 0.057 | **0.049** |
| 1024 | 0.081 | 0.102 | **0.077** |

**Score estimation.** Figure 2 (left) shows that DiScoFormer achieves much better MSE than KDE even in dimension 1 and especially in dimension 10. The model generalizes to GMMs with more modes than seen during training (right panel).

**Out-of-distribution generalization.** Despite training only on GMMs, the model generalizes to non-Gaussian distributions. We evaluate on the Laplace distribution (sharper peak, heavier tails) and the Student-$t$ distribution ($\nu = 3$, polynomial tail decay). Tables 2 and 3 show that DiScoFormer outperforms KDE on both. Performance can be further improved via test-time training (TTT) (Sun et al., 2020; Gandelsman et al., 2022) using the consistency loss $\mathcal{L}_{\mathrm{con}} = \frac{1}{n}\sum_{i=1}^{n}\|S(X)_i - \nabla_{x_i}\log T(X)_i\|_2^2$, where $\nabla \log T(X)$ is computed via automatic differentiation (Figure 8).

**Density estimation via SD-KDE.** Score-debiased KDE Epstein et al. (2025) reduces bias from $O(h^2)$ to $O(h^4)$ by moving samples along the score before applying KDE. Figure 3 shows that SD-KDE with our learned score outperforms plain KDE and Emp-SD-KDE.

**Entropy and Fisher information.** The model enables accurate estimation of differential entropy $H(f) = -\mathbb{E}_f \log f$ and Fisher information $I(f) = \mathbb{E}_f \|\nabla \log f\|_2^2$ from samples. In both $d = 1$ and $d = 10$, DiScoFormer outperforms KDE-based methods (see Appendix D.3 for detailed results).

**Plasma simulation.** The Landau equation, a Fokker–Planck-type PDE used in plasma physics, requires score estimates at each timestep—an ideal setting for our amortized model, which avoids retraining as the particle distribution evolves. We reproduce experiments from Ilin et al. (2025) using our pretrained model as a drop-in score oracle. Figure 4 shows that DiScoFormer matches analytic covariance evolution while KDE-based solvers struggle.

## 5 CONCLUSION

We introduced DiScoFormer, a permutation- and affine-equivariant Transformer that jointly estimates density and score from i.i.d. samples in a single forward pass. Unlike score matching, which learns

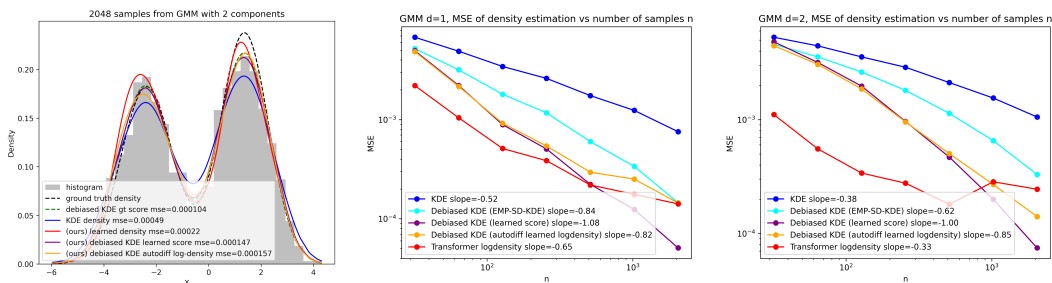

Figure 3: Left: 1D bimodal GMM – SD-KDE with learned score best approximates the density. Middle/Right: MSE of density estimation in 1D and 2D.

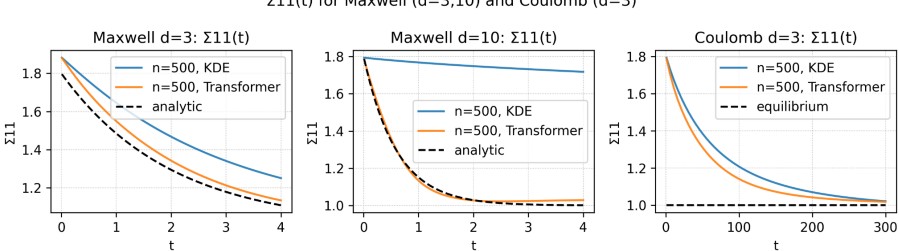

Figure 4: Landau equation: covariance $\Sigma_{1,1}(t)$ for Maxwell (left two) and Coulomb (right) collisions. DiScoFormer matches ground truth; KDE struggles.

$\mathbb{R}^d \to \mathbb{R}^d$ for one distribution, DiScoFormer learns a sequence-to-sequence operator that generalizes across distributions. We proved that self-attention recovers normalized KDE and confirmed empirically that individual heads learn multi-scale kernel-like behaviors. DiScoFormer outperforms KDE across dimensions and sample sizes and serves as a plug-in oracle for score-debiased KDE, entropy and Fisher information estimation, and Fokker–Planck-type PDEs.

**Limitations.** The current model is trained exclusively on GMMs with up to 10 modes and evaluated in dimensions $d \leq 10$. While GMMs are dense in the space of smooth densities and the model generalizes to non-Gaussian targets (Laplace, Student-$t$), broader evaluation on distributions with qualitatively different geometry (e.g., curved manifolds, heavy tails) and in higher dimensions is needed. The whitening layer provides exact equivariance under translation and scaling but only approximate rotation invariance via data augmentation. Systematic ablations of architectural choices (e.g., whitening vs. alternative equivariant designs) remain future work.

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

```python
def forward(self, X, Y):
    m = X.mean(0, keepdim=True)
    Xc, Yc = X - m, Y - m
    S = Xc.T @ Xc + eps * torch.eye(d)
    A = matrix_sqrt_inv(S)
    Xw, Yw = Xc @ A, Yc @ A
    dens, score = self._core(Xw, Yw)
    return dens * torch.det(A), score @ A.T
```

Figure 5: The forward pass implementing affine equivariance.

---

**Algorithm 1** GMM DataLoader

---

**Input:** $B, d, n_x, n_y, [k_{\min}, k_{\max}]$
**repeat**
    Sample $k \in \{k_{\min}, \ldots, k_{\max}\}$
    **for** $b = 1$ **to** $B$ **do**
        Sample two random $k$-component GMMs
        Sample $X_b, Y_b$ from each GMM
        Compute $f_{X_b}(y), \nabla \log f_{X_b}(y)$ for $y \in Y_b$
    **end for**
    **Output:** $(X, Y, f_X(Y), \nabla \log f_X(Y))$
**until** training stops

---

## A    IMPLEMENTATION DETAILS

The architecture is a permutation- and affine-equivariant Transformer. It consists of a linear embedding, four Transformer encoder layers (128-dimensional hidden size, 8 heads, GELU activation, pre-normalization), and a linear projection to the output dimension. No positional encodings are used, preserving permutation-equivariance. The model has about $800,000$ parameters. We use batch size 32, sample size $n = 2048$, dropout 0.1, and draw GMMs with 1–10 modes with means in $[-3, 3]^d$ and diagonal covariances in $[0, 1]^d$.

The training loss objective combines the mean squared errors (MSE) of the predicted density and score through a convex weighting:

$$\mathcal{L}_T = \frac{1}{n} \|T(X, Y) - f_X(Y)\|_2^2, \tag{1}$$

$$\mathcal{L}_S = \frac{1}{n} \|S(X, Y) - \nabla \log f_X(Y)\|_2^2 \tag{2}$$

$$\mathcal{L} = \alpha \mathcal{L}_T + (1 - \alpha) \mathcal{L}_S. \tag{3}$$

## B    ADDITIONAL ATTENTION VISUALIZATION

## C    PROOFS

**Proposition C.1** (Permutation and affine equivariance of density and score evaluation)**.** *Let $f : \mathbb{R}^d \to (0, \infty)$ be a density, and for*

$$X = \begin{pmatrix} x_1 \\ \vdots \\ x_n \end{pmatrix} \in \mathbb{R}^{n \times d}$$

*define*

$$T(X) := \begin{pmatrix} f(x_1) \\ \vdots \\ f(x_n) \end{pmatrix}, \qquad S(X) := \begin{pmatrix} \nabla \log f(x_1) \\ \vdots \\ \nabla \log f(x_n) \end{pmatrix}.$$

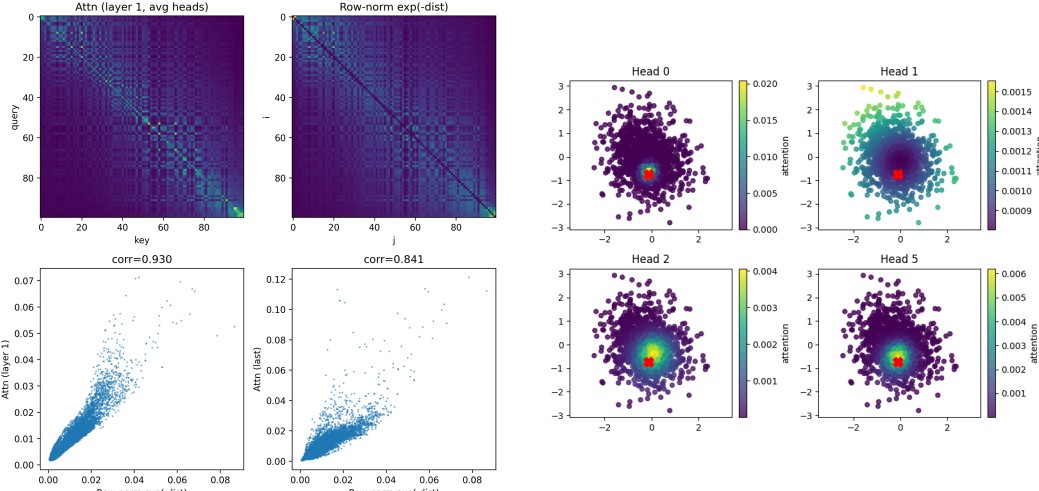

Figure 6: Left: Attention matrix and the normalized KDE matrix $D_{i,j} \propto e^{-\|x_i - x_j\|_2^2}$, with scatter plots showing very high agreement. Right: Attention scores of individual heads, demonstrating emergent head specialization.

*Let $P \in \mathbb{R}^{n \times n}$ be a permutation matrix, $A \in \mathbb{R}^{d \times d}$ be invertible, $\mu \in \mathbb{R}^d$, and $\mathbf{1} \in \mathbb{R}^n$ be the vector of ones. Then*

$$T\big(PXA + \mathbf{1}\mu^\top\big) = P\,|\det A|^{-1}\,T(X), \tag{4}$$

$$S\big(PXA + \mathbf{1}\mu^\top\big) = P\,A^{-1}\,S(X). \tag{5}$$

*Proof.* Write $X = (x_1^\top, \ldots, x_n^\top)^\top$ and let $\sigma$ be the permutation of $\{1, \ldots, n\}$ corresponding to $P$, i.e. $(PX)_{i\cdot} = x_{\sigma(i)}^\top$. Define

$$Y := PXA + \mathbf{1}\mu^\top \in \mathbb{R}^{n \times d}.$$

Then the $i$-th row of $Y$ is

$$y_i^\top = x_{\sigma(i)}^\top A + \mu^\top, \qquad \text{equivalently } y_i = A^\top x_{\sigma(i)} + \mu.$$

Consider the density obtained from $f$ by the affine change of variables $y = A^\top x + \mu$:

$$f^{(\mu,A)}(y) := |\det A|^{-1} f\big((A^\top)^{-1}(y - \mu)\big).$$

By construction,

$$f^{(\mu,A)}(y_i) = |\det A|^{-1} f\big(x_{\sigma(i)}\big).$$

Hence, by the definition of $T$,

$$T(Y) = \begin{pmatrix} f^{(\mu,A)}(y_1) \\ \vdots \\ f^{(\mu,A)}(y_n) \end{pmatrix} = \begin{pmatrix} |\det A|^{-1} f(x_{\sigma(1)}) \\ \vdots \\ |\det A|^{-1} f(x_{\sigma(n)}) \end{pmatrix} = P\,|\det A|^{-1}\,T(X),$$

which proves the first claim.

For the score, differentiate $\log f^{(\mu,A)}$:

$$\log f^{(\mu,A)}(y) = -\log|\det A| + \log f\big((A^\top)^{-1}(y - \mu)\big).$$

By the chain rule,

$$\nabla_y \log f^{(\mu,A)}(y) = \big((A^\top)^{-1}\big)^\top \nabla_x \log f(x)\big|_{x = (A^\top)^{-1}(y-\mu)} = A^{-1} \nabla \log f\big((A^\top)^{-1}(y - \mu)\big).$$

Evaluating at $y_i = A^\top x_{\sigma(i)} + \mu$ gives

$$\nabla_y \log f^{(\mu,A)}(y_i) = A^{-1} \nabla \log f(x_{\sigma(i)}).$$

Thus,

$$S(Y) = \begin{pmatrix} \nabla_y \log f^{(\mu,A)}(y_1) \\ \vdots \\ \nabla_y \log f^{(\mu,A)}(y_n) \end{pmatrix} = \begin{pmatrix} A^{-1}\nabla \log f(x_{\sigma(1)}) \\ \vdots \\ A^{-1}\nabla \log f(x_{\sigma(n)}) \end{pmatrix} = P\,A^{-1}\,S(X),$$

which proves the second claim.      □

**Proposition C.2** (Attention can compute KDE). *Let $X \in \mathbb{R}^{n \times d}$ be the normalized data matrix with $\|x_i\| = 1$ for all $i$. Consider a single self-attention head*

$$\text{Attention}(Q, K, V) = \text{Softmax}(QK^\top)\,V$$

*with*

$$Q = K = \tfrac{1}{h}X, \qquad V = I_n,$$

*for some bandwidth parameter $h > 0$. Then*

$$[\text{Attention}(Q, K, V)]_{ij} = \frac{\exp\left(-\frac{\|x_i - x_j\|^2}{2h^2}\right)}{\sum_{k=1}^n \exp\left(-\frac{\|x_i - x_k\|^2}{2h^2}\right)},$$

*i.e. the attention matrix coincides with the normalized Gaussian KDE kernel of bandwidth $h$.*

*Proof.* With $Q = K = X/h$, we have $QK^\top = XX^\top/h^2$, so

$$\text{Softmax}(QK^\top)_{ij} = \frac{\exp(x_i^\top x_j/h^2)}{\sum_k \exp(x_i^\top x_k/h^2)}.$$

Because all $\|x_i\| = 1$,

$$x_i^\top x_j = 1 - \tfrac{1}{2}\|x_i - x_j\|^2.$$

Substituting and simplifying gives

$$\exp(x_i^\top x_j/h^2) = e^{1/h^2} \exp(-\|x_i - x_j\|^2/2h^2),$$

and the common factor $e^{1/h^2}$ cancels in the row-wise softmax normalization. Hence

$$\text{Softmax}(QK^\top)_{ij} = \frac{\exp(-\|x_i - x_j\|^2/2h^2)}{\sum_k \exp(-\|x_i - x_k\|^2/2h^2)}.$$

Since $V = I_n$, multiplying by $V$ leaves this matrix unchanged, and the claim follows.      □

## D    Additional Experiment Details

### D.1    Score Estimation

The KDE score approximation is computed using

$$k(x_i, x_j) = \exp\left(-\frac{1}{2}\sum_{\ell=1}^d \frac{(x_i^\ell - x_j^\ell)^2}{h_\ell^2}\right), \tag{6}$$

$$\widehat{s}(x_i) = \frac{1}{h^2} \odot \left(\frac{\sum_j k(x_i, x_j)x_j}{\sum_j k(x_i, x_j)} - x_i\right), \tag{7}$$

where $h_\ell = \sigma_\ell n^{-1/(d+4)}$ is Silverman's rule Silverman (1986) applied per coordinate with $\sigma_\ell$ the sample standard deviation along dimension $\ell$, and $\odot$ denotes element-wise multiplication.

### D.2    Out-of-Distribution Generalization

Quantitative results for the Laplace and Student-$t$ out-of-distribution experiments are in Tables 2 and 3 in the main text. Figure 7 provides a qualitative comparison of score fields, and Figure 8 shows MSE as a function of sample size on the Laplace distribution, including test-time training improvements.

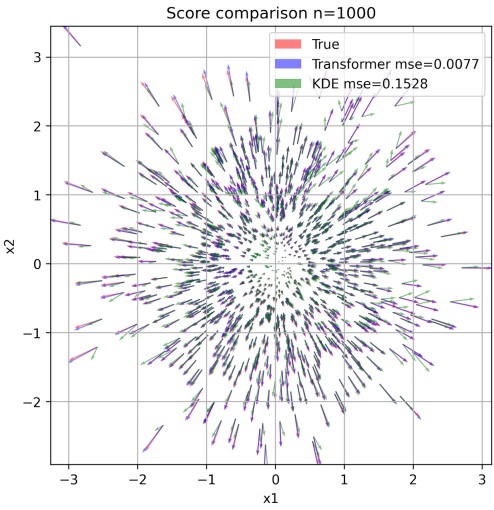

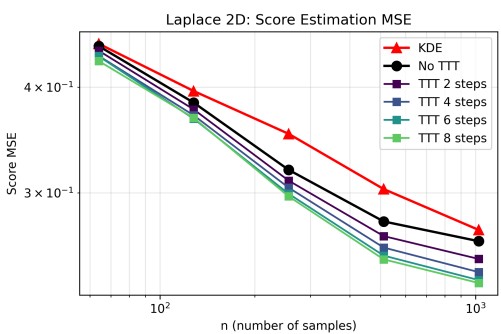

Figure 8: MSE of score estimation on the Laplace distribution. Test-time training (TTT) improves the out-of-distribution generalization.

Figure 7: Score estimation comparison between Silverman KDE and our transformer model. The transformer is more accurate, especially in the sparse regions.

### D.3 ENTROPY AND FISHER INFORMATION

Differential entropy $H(f) = -\mathbb{E}_f \log f$ and its gradient, Fisher information $I(f) = \mathbb{E}_f \|\nabla \log f\|_2^2$, are fundamental statistical quantities. We use the discretizations

$$H(f) \approx \frac{1}{n} \sum_{i=1}^{n} \log f(x_i) \approx \frac{1}{n} \sum_{i=1}^{n} \log T(X)_i \tag{8}$$

$$I(f) \approx \frac{1}{n} \sum_{i=1}^{n} \|\nabla \log f(x_i)\|^2 \approx \frac{1}{n} \sum_{i=1}^{n} \|S(X)_i\|^2 \tag{9}$$

to estimate $H(f)$ and $I(f)$ using only an iid sample of $f$. Figure 9 demonstrates that already in dimension 1 both the score model $S$ and the density model $T$ outperform Silverman KDE. In dimension 10 the learned density performs much better for entropy estimation and the learned score performs much better for Fisher information estimation.

### E RELATIVE FISHER INFORMATION

The Relative Fisher Information (RFI) measures how two densities differ in their local geometry, comparing their score fields rather than their probability mass. Unlike the KL divergence, which captures global differences, RFI focuses on the mismatch between gradients of the log-densities, offering a more sensitive criterion for evaluating score-based models and generative estimators.

Formally, for two densities $f$ and $g$ on $\mathbb{R}^d$, the relative Fisher information of $g$ with respect to $f$ is defined as

$$\mathrm{I}(g\|f) = \mathbb{E}_{x\sim f}\left[\|\nabla_x \log g(x) - \nabla_x \log f(x)\|_2^2\right],$$

and can be approximated by the Monte Carlo estimate on samples $x_i \sim f$ with

$$\mathrm{I}(g\|f) \approx \frac{1}{n} \sum_{i=1}^{n} \|\nabla \log g(x_i) - \nabla \log f(x_i)\|_2^2.$$

Because our model outputs the scores $\nabla \log g(x_i)$ using samples $y_i$ from $g$, we can compute relative Fisher information between two distributions using their samples, as visualized in Figure 10.

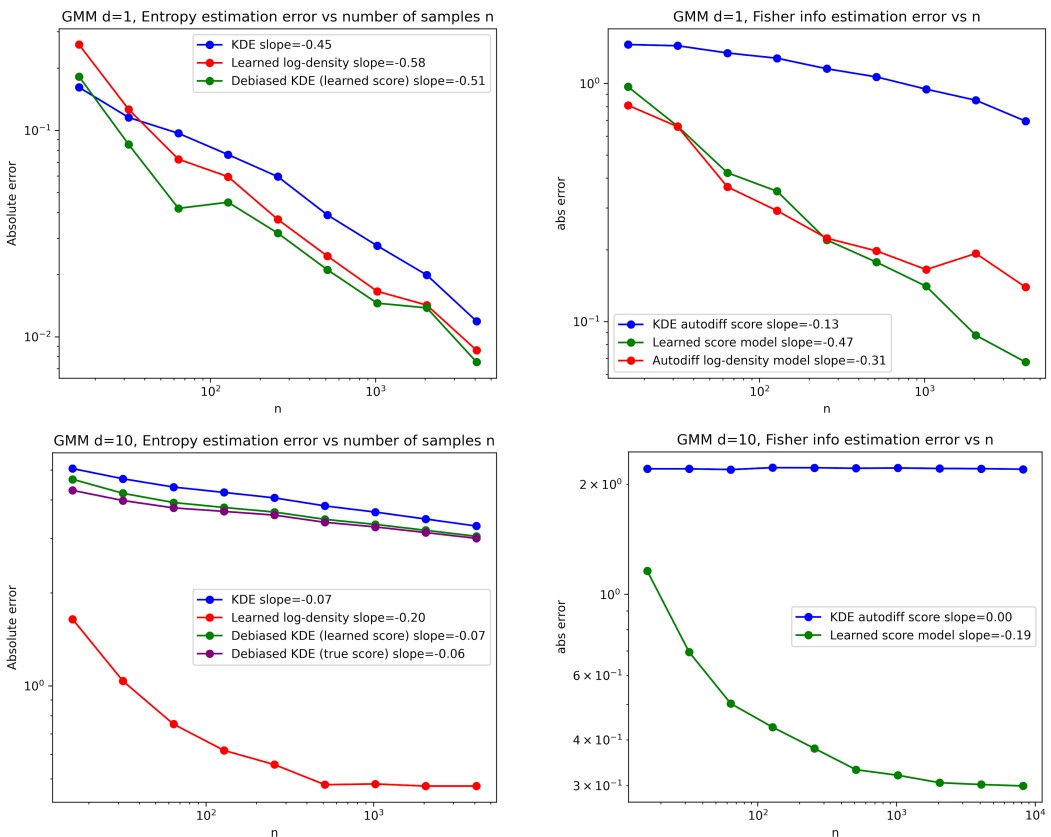

Figure 9: Comparison between the transformer model (learned) and Silverman KDE and score-debiased KDE for estimation of differential entropy $H(f)$ and Fisher Information $I(f)$. Transformer's MSE is lower than that of the KDE approximation, even in dimension 1.

# F Attention visualization

To interpret the outputs of the Transformer model we visualize the attention of the eight individual heads in layer 0 in two different ways. First, in Figure 11 we show the full attention matrices as heatmaps. We choose the particle ordering so that nearby particles are close in the ordering. Second, in Figure 12 we choose a random query point, marked with a red x, and color other points according to the strength of the query point's attention. We clearly observe an emergent behavior – heads specialize in different tasks. Head 1 specialized to look at far-away points, whereas Heads 0, 2, and 5 specialized to look at close- and mid-range interactions. Finally, Heads 3, 4, 6, and 7 specialized to look in specific directions. This emergent behavior further links the multi-head Transformer to kernel-based methods, but with multi-scale learned kernels.

# G Comparison with the Score Matching Loss

The Hyvrinen score matching loss Hyvärinen (2005) estimates the data score without access to the true density by minimizing

$$\mathcal{L}_{\mathrm{SM}}(\theta) = \mathbb{E}_{x \sim f(x)}\left[\| s_\theta(x) \|^2 + 2 \nabla_x \cdot s_\theta(x)\right],$$

which corresponds to the Fisher divergence $\mathbb{E}_f[\|s_\theta(x) - \nabla_x \log f(x)\|^2]$ up to a constant.

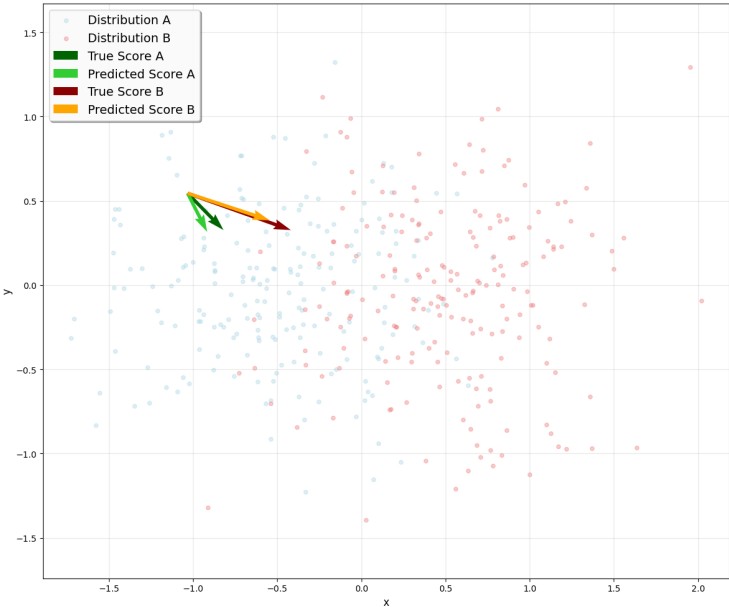

Figure 10: Computing relative Fisher information. Our model predicts $\nabla \log g(x_i)$ at query points $x_i$ via cross-attention with the samples $y_i \sim g$. These predicted scores can be used to estimate the relative Fisher information.

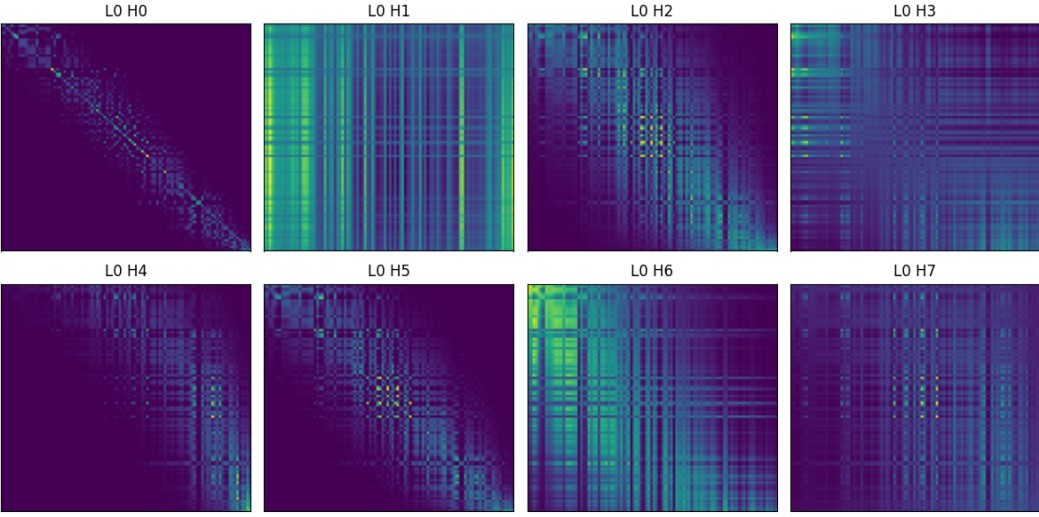

Figure 11: We visualize attention of the eight individual heads of layer 0 as a heatmap. Yellow color denotes higher attention weight and blue is lower. We choose the particle ordering so that nearby particles are close in the ordering. Heads 0, 2, and 5 specialize on nearby points, heads 1 and 6 specialize on far-away points, whereas heads 3, 4, and 7 attend in specific directions.

To avoid explicitly computing the divergence term, one can apply a finite-difference denoising trick, replacing $\nabla \cdot s_\theta(x)$ with a stochastic estimator

$$\nabla \cdot s_\theta(x) \approx \mathbb{E}_{z \sim \mathcal{N}(0,I)} \left[ \frac{z^\top \left( s_\theta(x + \alpha z) - s_\theta(x - \alpha z) \right)}{2\alpha} \right],$$

yielding a practical divergence-free approximation used in many implementations of score matching.

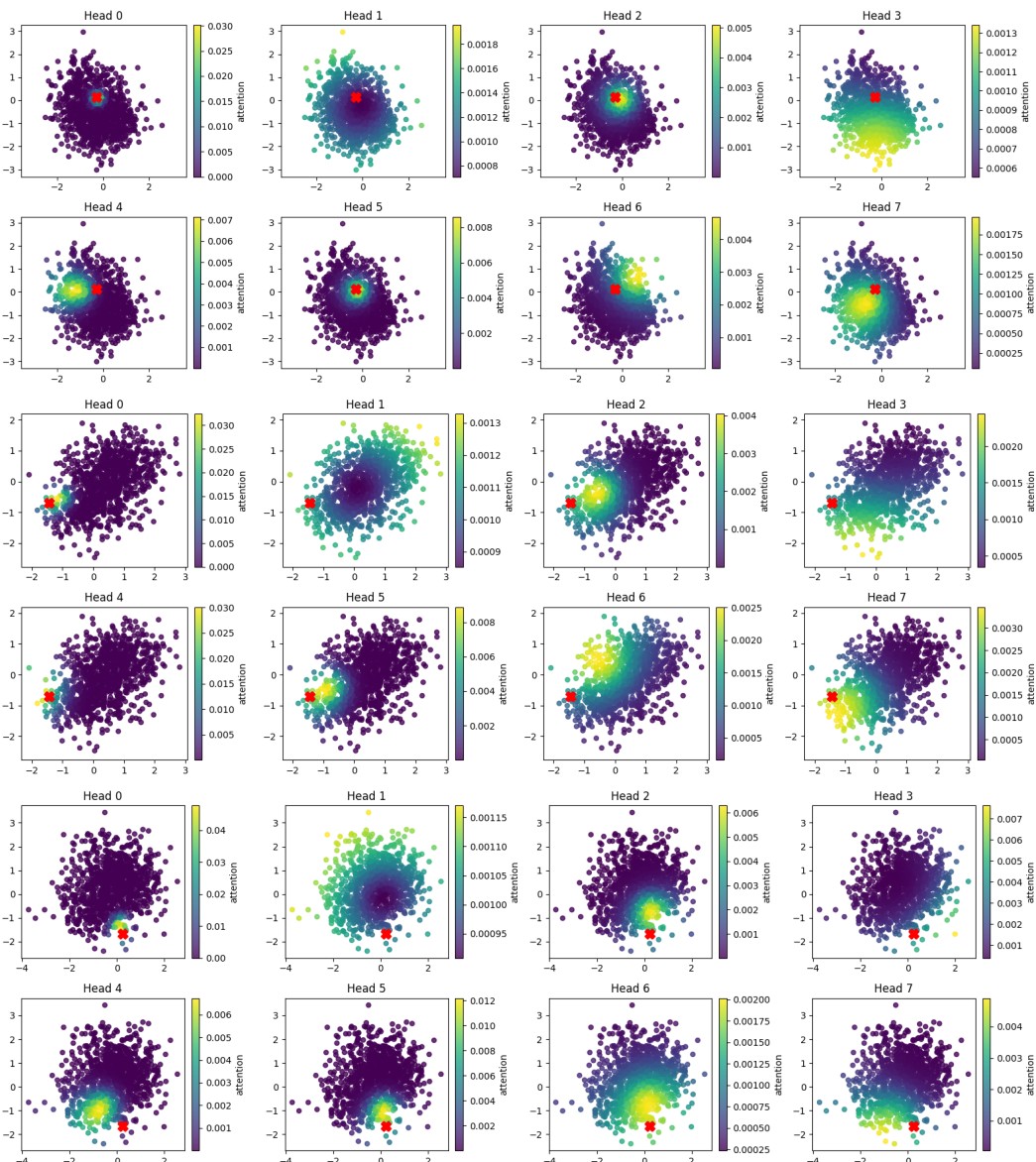

Figure 12: We visualize attention of the eight individual heads of layer 0 as a scatter plot. We choose a random query point, marked with a red x. Yellow color denotes higher attention weight and blue is lower. Heads 0, 2, and 5 specialize on nearby points, head 1 specializes on far-away points, whereas heads 3, 4, 6, and 7 attend in specific directions.

In Figure 13 we compare the performance of the score matching loss compared to the proposed Transformer model. The Transformer model is superior in two respects. First, it does not need to be retrained on new samples. Second, the score matching loss is easy to under- or over-fit, as demonstrated in the figure by training for 10, 100 and 1000 steps.

## H  RUNTIME COMPARISON

While KDE is known to fail in moderate and high dimensions, it is thought to be very fast. We compare the wall-clock runtime of KDE score estimation as in equation , and the Transformer model we propose. The Transformer model contains $800,000$ learned parameters. We use both KDE and the Transformer model on a single L40S GPU with 48Gb of memory. Both methods

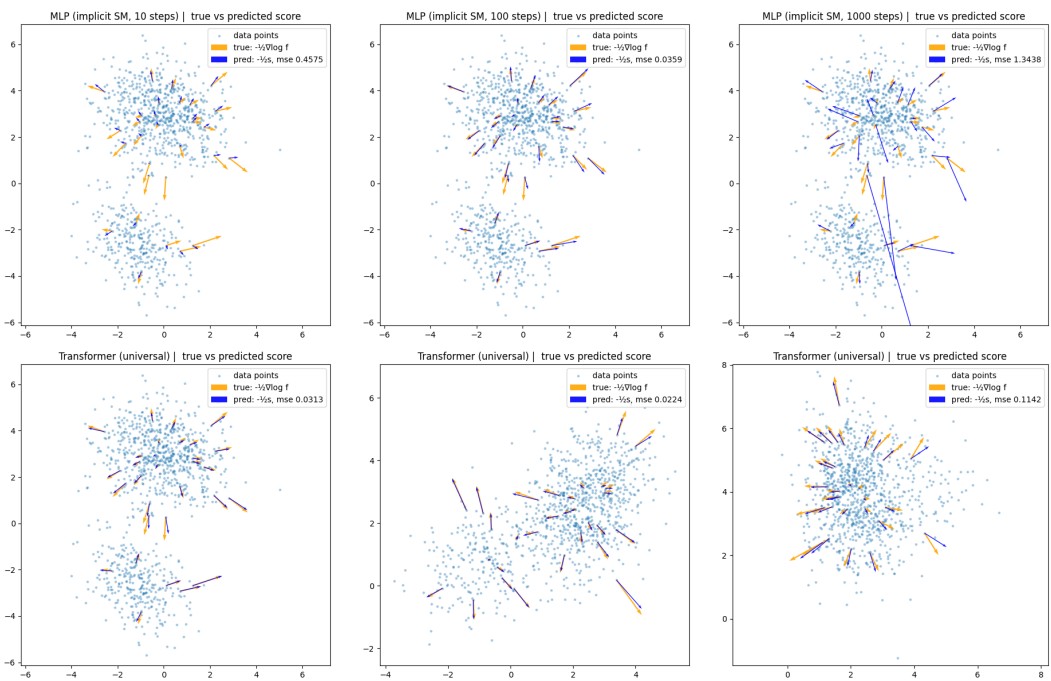

Figure 13: Comparison of sliced score matching and our transformer model. The transformer model can be used without retraining and does not suffer from overfitting. We plot the negated score for ease of visualization.

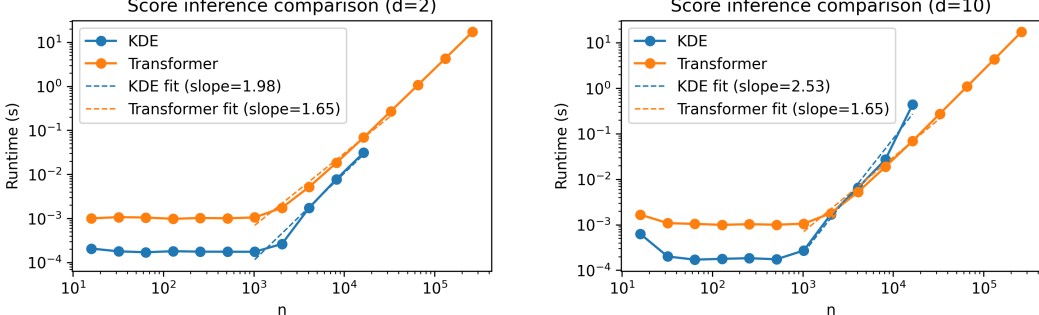

Figure 14: Runtime comparison between KDE and the Transformer model in 2 and 10 dimensions. Both are $O(n^2)$ asymptotically, but empirically the Transformer scales better and has improved memory efficiency. KDE encounters an OOM error at $n = 2^{15}$.

are $O(n^2)$ asymptotically due to pairwise computations (kernel evaluations for KDE, attention for Transformers). However, in practice the observed scaling differs: Figure 14 shows that while KDE is faster for small sample size $n \leq 2048$, it becomes slower after this point, especially in higher dimension. Additionally, the naive KDE implementation runs out of GPU memory after $n = 32,768$, while the Transformer benefits from the highly optimized attention kernels in modern deep learning frameworks, which are difficult to match with custom KDE implementations in practice.

