# OpenReview forum: "DiScoFormer: Plug-In Density and Score Estimation with Transformers"
_ICLR.cc/2026/Workshop/GRaM — ICLR 2026 Workshop GRaM Poster_

### Official Review · Reviewer_WKhN · 2026-02-08

**Rating:** 2
**Confidence:** 4

**Review:**

This paper presents DiScoFormer, a transformer for predicting the score and density of general distributions. Some theoretical results show the equivariances of score and density by affine transformations and that the attention mechanism can compute KDE in theory. There are experiments testing the model on GMMs and other simple distributions.

- While it's true KDE is equivariant to affine transformations, not all probability densities are related by affine transformations. The "train-once, infer-anywhere" claim seems to be greatly overstated and there's no discussion/justification why this transformation is sufficient for modelling general densities.
- If I understand correctly the architecture is a vanilla transformer with a whitening step canonicalizing the density. This design decision isn't ablated against any sensible alternative eg. vanilla transformer without canonicalization, equivariant transformer.
- It's not surprising a transformer can do density estimation for Gaussian mixtures. The work would be much more compelling with experiments on complex real-world distributions.
- The presentation and writing can be greatly improved. Plots should be cleaned up and missing relevant discussion on the limitations and scope of generalization should be added. It's necessary to read the appendix to understand core parts of the work.

**Pmlr Suitability:**

NA

---

### Official Review · Reviewer_YTj7 · 2026-02-09
**Promising idea but severely limited scope and missing critical baselines**

**Rating:** 4
**Confidence:** 4

**Review:**

## Strengths
* **Novel paradigm**: Train-once approach avoids per-distribution retraining required by neural score matching
* **Solid theory**: Proposition 3.2 elegantly shows self-attention can recover normalized KDE
* **Interpretable attention**: Emergent multi-scale head specialization (Figures 11-12) provides insights
* **Comprehensive applications**: Demonstrates utility for SD-KDE, Fisher information, entropy, and Fokker-Planck PDEs
* **Strong performance within scope**: Consistent improvements over KDE on tested distributions

## Critical Weaknesses
* **GMM-only training**: Model trained exclusively on GMMs (1-10 modes, diagonal covariances)—this fundamentally undermines "infer-anywhere" claims. Title/abstract should explicitly state this limitation
* **Inadequate OOD evaluation**: Only tested on Laplace and Student-t (both unimodal, symmetric). Missing: heavy-tailed, skewed, manifold-supported, non-GMM multimodal distributions. Test-time training needed for OOD (Table 3) contradicts "train-once" claim
* **Missing score matching baselines**: No quantitative comparison with denoising/sliced score matching or modern neural methods. This is the primary baseline for score estimation—comparison only to KDE is insufficient
* **Limited dimensionality**: All experiments $d \leq 10$. Need evidence in $d = 50, 100$ to validate "curse of dimensionality" claims
* **No architecture ablations**: Whitening + data augmentation vs. alternatives not tested. Rotation equivariance only approximate ($5 \times 10^{-4}$ error)

## Moderate Issues
* **Incomplete details**: Missing $\alpha$ (Eq. 3), learning rate schedule, epochs, numerical stability of matrix inverse
* **Weak evaluation metrics**: Only relative $\ell_2$ error—should include KL, Wasserstein, MMD, generative modeling metrics
* **Theory-practice gap**: Proposition 3.2 assumes $\|x_i\| = 1$ and specific $Q, K, V$ never used in practice
* **Cross-attention unexplained**: Mentioned but never detailed or evaluated
* **Overselling**: "Universal," "infer-anywhere" language not supported. Figure 2 calls more GMM modes "OOD generalization"

## Missing Experiments
* Quantitative score matching comparisons (denoising, sliced)
* Systematic OOD evaluation (Cauchy, power-law, skewed, manifold distributions)
* Higher dimensions ($d = 20, 50, 100$)
* Ablations (equivariance components, loss weighting, architecture)
* Real-world data applications
* Quantitative Landau equation metrics (Figure 4 only visual)

## Questions
1. What justifies that GMM training generalizes to arbitrary distributions?
2. How does this compare quantitatively to sliced/denoising score matching?
3. Can you show performance in $d > 10$ and on truly OOD distribution families?
4. What is $\alpha$ in Eq. 3 and what ablations were performed?

**Pmlr Suitability:**

NA

---

### Official Review · Reviewer_n1UB · 2026-02-21
**Review of DiScoFormer**

**Rating:** 5
**Confidence:** 2

**Review:**

This paper proposes DiScoFormer, a Set-Transformer-style model that takes iid samples and predicts density and score at query points. It enforces (approximate) affine equivariance via a whitening/canonicalization step and argues attention can represent (Gaussian) KDE weights in theory. Training and evaluation are mostly on synthetic GMMs (with closed-form density/score supervision), plus a couple of simple OOD distributions and a few plug-in demos.
•	The train-once to infer elsewhere feels a little overstated. While whitening gives affine equivariance as a symmetry property, there is no real justification that this is sufficient for general densities, and rotation handling is only approximate/augmented.
•	The model is essentially a vanilla transformer with canonicalization, yet ablations against obvious baselines (no whitening, alternative equivariant designs) are limited, so it is difficult to credit the specific architectural choices.
•	Empirically it is not surprising a transformer can learn density/score on GMMs; the work would be much more convincing with tests on more difficult real-world distributions and stronger density-estimation baselines.

**Pmlr Suitability:**

NA

---

### Official Review · Reviewer_h5MN · 2026-02-24
**DiScoFormer - needs further empirical evaluation**

**Rating:** 6
**Confidence:** 3

**Review:**

The paper proposes a “one-shot” method for estimating both a density and its score from i.i.d. samples. It uses a Transformer architecture and builds in the required symmetries—permutation invariance/equivariance by omitting positional encodings, and affine equivariance through a whitening-based normalization layer with the appropriate output transformation.

The authors also show that, for a specific parameterization, Transformer attention has the capacity to implement kernel-based smoothing, providing a conceptual link to KDE.

Empirically, the authors train and evaluate on Gaussian mixture models (where density and score are available in closed form) and report improved score MSE over KDE up to $d=10$. They further demonstrate downstream use as a plug-in score oracle for score-debiased KDE, Fisher information estimation, and Fokker–Planck / probability-flow ODE applications.

Overall, this is an interesting contribution, and my inclination is marginal accept. I like the problem framing and the symmetry-aware amortized operator, and I view the KDE–attention connection as a nice supporting result. However, I find the main “infer-anywhere” generalization claim overstated given the scope of the OOD evaluation provided in this short paper (however I acknowledge that this may be due to limited space).

---
### Strengths

- I like the symmetry-aware design: permutation equivariance and affine equivariance are explicitly built into the proposed architecture.
- The attention–KDE result is a clean argument that supports the claim that Transformers are  adaptive kernel smoothers, and to the best of my knowledge this argument is new.



---
### Weaknesses


- My main issue with the framework is that the authors compare the score estimation only against a KDE estimator, while there are much more efficient and accurate frameworks for this. For example see Steins method[1, 7,9], Kernel based GP-inspired methods [2,5], Logarithmic gradient estimation from denoising auto-encoders [[Raphan and Simoncelli,
2011; Vincent, 2011; Vincent, Larochelle, Bengio, et al., 2008], Sliced Score Matching [6], variational methods [3], spectral methods [4] and other (see the review in [8]). Thus given the breadth of the existing methods I would expect a more extensive comparison with existing approaches, but for a workshop paper this can be improved until the workshop presentation. Even for KDE I would expect a comparison with an adaptive KDE variant.

- While I understand the incentive on training on Gaussian mixture models, I would also expect comparisons of more challenging distributions with other baseline methods (see for instance [8]). But as said before for a workshop paper this is still ok.

- I am not entirely convinced with the "Infer-anywhere" generalization claim the authors make and I feel (at least in the limited space of this short paper) this is not convincingly established by training only on GMMs. So thus far I think the empirical evaluation doesn’t justify broad distribution-agnostic claims.

- What experiments would make convinced:

     -  Test on non-GMM families with qualitatively different geometry: strongly correlated anisotropic densities, curved “banana” distributions, rings, mixtures with widely varying scales.
     -  Fairer baseline set

---
### Minor

- To make Figure 1 more clear I would suggest to add some transparency to the points indicating the data samples to make the arrows and indicating the score stand out more.The current version of the figure is too crowded with information/colors.



---
### References

[1] Stein, C. M. Estimation of the mean of a multivariate
normal distribution. The annals of Statistics, pp. 1135–
1151, 1981.


[2] Sriperumbudur, B., Fukumizu, K., Gretton, A., Hyväri-
nen, A., and Kumar, R. Density estimation in infinite
dimensional exponential families. Journal of Machine
Learning Research, 18(57):1–59, 2017


[3] Maoutsa, D., Reich, S., & Opper, M. (2020). Interacting particle solutions of fokker–planck equations through gradient–log–density estimation. Entropy, 22(8), 802.


[4] Shi, J., Sun, S., and Zhu, J. A spectral approach to gra-
dient estimation for implicit distributions. In Proceed-
ings of the 35th International Conference on Machine
Learning, pp. 4651–4660, 2018.

[5] Sutherland, D., Strathmann, H., Arbel, M., and Gretton,
A. Efficient and principled score estimation with Nys-
tröm kernel exponential families. In Proceedings of
the Twenty-First International Conference on Artificial
Intelligence and Statistics, pp. 652–660, 2018.

[6] Song, Y., Garg, S., Shi, J., & Ermon, S. (2020, August). Sliced score matching: A scalable approach to density and score estimation. In Uncertainty in artificial intelligence (pp. 574-584). PMLR.

[7] Li, Yingzhen and Richard E Turner. “Gradient estimators
for implicit models”. International Conference on Learning Representations (2018)

[8] Zhou, Y., Shi, J., & Zhu, J. (2020, November). Nonparametric score estimators. In International Conference on Machine Learning (pp. 11513-11522). PMLR.

[9] Korba, A., Aubin-Frankowski, P. C., Majewski, S., & Ablin, P. (2021, July). Kernel stein discrepancy descent. In International Conference on Machine Learning (pp. 5719-5730). PMLR.

**Pmlr Suitability:**

NA

---

### Meta-Review · Area_Chair_6bsB · 2026-02-25

**Decision:**

Accept

**Metareview:**

The reviewers find some ideas in the paper to be interesting with some novel arguments. The main weakness seems to be the "train-once infer-anywhere" claim which reviewers do not find convincing evidence for and therefore may need to be softened. The reviewers agree more experiments would be useful, which would also help to substantiate the claim. However, due to the novel ideas, I recommend accepting this paper but suggest that the authors take some of the feedback from the reviewers into account in order to improve the paper.

**Relevance To Proceedings:**

Tiny paper — does not apply

**Relevance To Workshop:**

Yes — suitable for GRaM

---

### Decision · Program_Chairs · 2026-03-02

Accept (Poster)